# Human induced pluripotent stem cell-derived cardiomyocytes to study inflammation-induced aberrant calcium transient

Yuki Tatekoshi[1†], Chunlei Chen[1†], Jason Solomon Shapiro[1], Hsiang-Chun Chang[1], Malorie Blancard[2], Davi M Lyra-Leite[2], Paul W Burridge[2], Matthew Feinstein[3], Richard D'Aquila[3], Priscilla Hsue[4], Hossein Ardehali[1,2,3]*

[1]Feinberg Cardiovascular and Renal Research Institute, Northwestern University, Chicago, United States; [2]Department of Pharmacology, Northwestern University, Chicago, United States; [3]Department of Medicine, Northwestern University, Chicago, United States; [4]Department of Medicine, University of California, San Francisco, San Francisco, United States

**\*For correspondence:**
h-ardehali@northwestern.edu

[†]These authors contributed equally to this work

**Abstract** Heart failure with preserved ejection fraction (HFpEF) is commonly found in persons living with HIV (PLWH) even when antiretroviral therapy suppresses HIV viremia. However, studying this condition has been challenging because an appropriate animal model is not available. In this article, we studied calcium transient in human induced pluripotent stem cell-derived cardiomyocytes (hiPSC-CMs) in culture to simulate the cardiomyocyte relaxation defect noted in PLWH and HFpEF and assess whether various drugs have an effect. We show that treatment of hiPSC-CMs with inflammatory cytokines (such as interferon-γ or TNF-α) impairs their $Ca^{2+}$ uptake into sarcoplasmic reticulum and that SGLT2 inhibitors, clinically proven as effective for HFpEF, reverse this effect. Additionally, treatment with mitochondrial antioxidants (like mito-Tempo) and certain antiretrovirals resulted in the reversal of the effects of these cytokines on calcium transient. Finally, incubation of hiPSC-CMs with serum from HIV patients with and without diastolic dysfunction did not alter their $Ca^{2+}$-decay time, indicating that the exposure to the serum of these patients is not sufficient to induce the decrease in $Ca^{2+}$ uptake in vitro. Together, our results indicate that hiPSC-CMs can be used as a model to study molecular mechanisms of inflammation-mediated abnormal cardiomyocyte relaxation and screen for potential new interventions.

## eLife assessment

This **useful** study focuses on heart failure with preserved ejection fraction (HFpFE), common in patients with HIV. Researchers used induced human pluripotent stem cell-derived cardiomyocytes (hiPSC-CMs) to stimulate (HEFpEF) and found that inflammatory cytokines alter Ca2+ transients. SGLT2 inhibitors and mitochondrial antioxidants reversed this effect. While the study is **incomplete** and preliminary, its strength lies in introducing hiPSC-CMs as a tool for investigating HFpEF mechanisms. A major weakness was found to be limited functional assessment relevant to HFpEF.

## Introduction

Heart failure (HF) with preserved ejection fraction (HFpEF) refers to a form of HF that is associated with diastolic dysfunction (DD). On echocardiography, HFpEF is associated with a greater than 50% of

blood in the left ventricle (LV) that is pumped with each beat but the LV displays impaired relaxation and increased stiffness during the filling phase of cardiac cycle. Echocardiographic evidence of DD is associated with higher all-cause mortality even in asymptomatic patients (*Kardys et al., 2009*). The pathophysiology underlying DD is known to be an increase in the stiffness of the LV, although there are limited ways to study the mechanisms and interventions. Studies have shown that patients with HFpEF have multiple comorbidities, such as obesity, hypertension, renal dysfunction, and diabetes (*Dhingra et al., 2014*; *Haass et al., 2011*), and that it is associated with aging (*Xu and Daimon, 2016*). It is also proposed that these comorbid conditions each lead to increased systemic inflammation, which then causes myocardial remodeling and fibrosis (*Dhingra et al., 2014*; *Cheng et al., 2013*; *Sanders-van Wijk et al., 2015*; *Santhanakrishnan et al., 2012*).

There are few options for the treatment of HFpEF, despite the increasing incidence of this disease in HIV-uninfected and -infected patients. The current treatments relieve symptoms and target comorbid conditions, such as hypertension, diabetes, coronary artery disease, hyperlipidemia, and atrial fibrillation. Although beta blockers, ACE inhibitors, angiotensin receptor blockers, and cardiac resynchronization therapy improve outcome in patients with HF with reduced ejection fraction, clinical benefits have scarcely been shown with any of these drugs in HFpEF. The only drugs that have shown clinical benefit in rigorous trials are antagonists of the mineralocorticoid receptor (such as spironolactone) (*Pfeffer and Braunwald, 2016*), inhibitors of the sodium glucose co-transporter 2 (SGLT2, such as empagliflozin) (*Anker et al., 2021*), and glucagon-like peptide 1 analog (such as semaglutide) (*Kosiborod et al., 2023*).

Persons living with HIV (PLWH) have a higher prevalence of HFpEF compared to uninfected individuals (*Meng et al., 2002*; *Schuster et al., 2008*). Recent studies have shown that LV mass index and DD are each significantly worse in PLWH with HFpEF than among persons living without HIV with HFpEF, despite similar EF. These features were associated with lower nadir $CD4^+$ T-cell count, suggesting that this process could be due to the level of immunodeficiency and/or duration of untreated infection (*Hsue et al., 2010*). Several studies attempted to determine the mechanism for increased HFpEF in PLWH. Administration of HIV gp120 resulted in DD after adrenergic stimulation in rats (*Berzingi et al., 2009*) and caused negative inotropic effects in adult rat ventricular myocytes (*Kan et al., 2004*; *Yuan et al., 2008*). Macaques infected with simian immunodeficiency virus (SIV) display DD, which was associated with the degree of myocardial SIV viral load. Inhibition of CCR5, a co-receptor with CD4 for HIV gp120-mediated entry into cells, in the SIV-infected rhesus macaque model of DD preserved cardiac diastolic function (*Kelly et al., 2014*).

Untreated HIV infection is associated with circulating inflammatory cytokines and chemokines (*Katsikis et al., 2011*), and many remain elevated even when HIV viremia is well-controlled by antiretroviral therapy (ART) (*Keating et al., 2011*; *Ramirez et al., 2014*). For instance, elevated IFN-γ is observed during acute HIV viremia and remained elevated despite highly active ART (HAART) (*Roff et al., 2014*). Higher IFN-γ is also associated with impaired CD4 recovery in HIV after the initiation of HAART (*Watanabe et al., 2019*). Additionally, while the immune system resets itself after the regression of bacterial or viral infection, it remains hypersensitive when ART suppresses viremia in chronic HIV infection. Of note, monocytes from PLWH treated with viremia suppressing ART demonstrated increased production of TNF-α and IL-6 after stimulation with lipopolysaccharide (*Amirayan-Chevillard et al., 2000*), indicating the presence of immune system rewiring during HIV infection that is not reversed with effective ART. Therefore, cardiomyocytes in PLWH are exposed to higher circulating inflammatory cytokines at baseline despite effective ART, and with further elevation of these inflammatory cytokines after concomitant infections. Among these cytokines, TNF-α is associated with mortality (*Wada et al., 2016*) and severe coronary stenosis (greater than 70% stenosis) *Bahrami et al., 2016* in PLWH.

There are limited studies to address the molecular basis of cardiomyocyte dysfunction in HIV-mediated HFpEF or the effectiveness of various drugs. Additionally, there are currently no models using human heart cells and HIV. These major limitations raise the critical need for a novel approach to study this complicated disease. Human induced pluripotent stem cell (hiPSC)-derived cardiomyocytes (hiPSC-CM) represent a novel technology that has been successfully applied to understanding basic mechanisms of several cardiovascular diseases, including long QT (*Itzhaki et al., 2011*), LEOPARD (*Carvajal-Vergara et al., 2010*) and Timothy (*Yazawa et al., 2011*) syndromes, dilated cardiomyopathy (CM) (*Sun et al., 2012*; *Lan et al., 2013*), and hypertrophic CM (*Lan et al., 2013*). They are

also used for efficacy and toxicity screening of drugs (*Liang et al., 2013*; *Navarrete et al., 2013*), leading to enhanced understanding of cellular mechanisms at the single cardiomyocyte level, not previously possible using human cells. Furthermore, hiPSC-CMs have been used to study a model of viral myocarditis (i.e., coxsackie virus) and predict the efficacy of antiviral drug therapies (*Sharma et al., 2014*), indicating that these cells can be used to study HF due to viral infections. In this article, we show that hiPSC-CMs can be used as a model for the assessment of calcium transient in response to inflammatory cytokines in vitro and to assess the effects of various drugs on their relaxation parameters. Given the major limitations in the field of HIV-mediated cardiac dysfunction, this model will allow investigators to study the mechanism of cardiac DD in inflammatory condition in vitro and assess the effectiveness of various drugs to reverse impaired relaxation noted with systemic inflammation.

## Results

### TNF-α and IFN-γ impair relaxation in hiPSC-CMs without causing cellular damage

We first generated hiPSCs and differentiated into cardiomyocytes using the protocols previously published by our groups (*Burridge et al., 2015*). Successful production of hiPSC-CMs was confirmed by the assessment of their morphology and calcium transient property (*Figure 1—figure supplement 1A and B*, *Figure 1—video 1*). We then assessed whether treatment with cytokines would have detrimental effects on these cells. Because DD has been associated with mitochondrial dysfunction (*Kumar et al., 2019*), we first measure whether cytokine treatment results in changes in mitochondrial

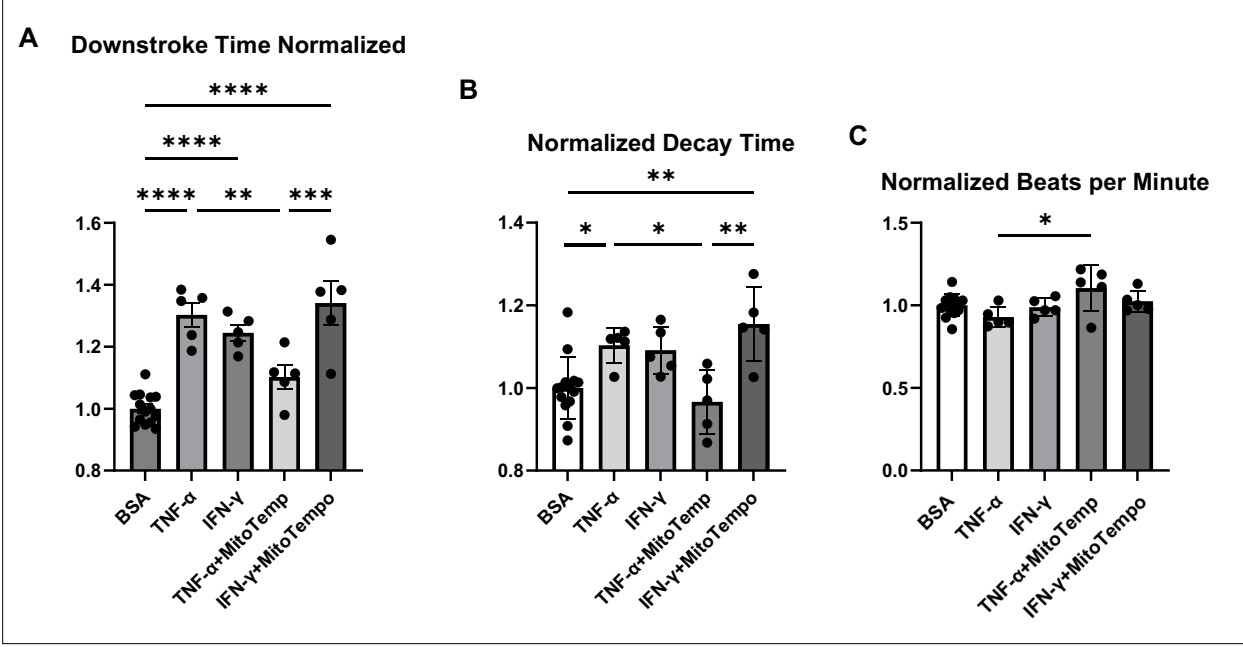

**Figure 1.** Analyses of calcium transient in hiPSC-CMs treated with cytokines and/or MitoTempo. Normalized downstroke time (**A**), decay time (**B**), and beating rate as assessed by beats per minute (**C**) in human induced pluripotent stem cell-derived cardiomyocytes (hiPSC-CM) treated with bovine serum albumin (BSA), (control), TNF-α, IFN-γ, TNF-α plus mito-Tempo, and IFN-γ plus mito-Tempo. N = 5–14. Data were analyzed by ordinary one-way ANOVA and post hoc Tukey's multiple-comparison test. Bars represent group mean, and error bars indicate standard error of the mean.

The online version of this article includes the following video, source data, and figure supplement(s) for figure 1:

**Source data 1.** Analyses of calcium transient in hiPSC-CMs treated with cytokines and/or MitoTempo.

**Figure supplement 1.** A representative calcium transient curve in hiPSC-CMs and their mitochondrial potential.

**Figure supplement 1—source data 1.** A epresentative image of hiPSC-CMs, a calcium transient curve, and a relative dF/dt curve.

**Figure supplement 1—source data 2.** Relative TMRE fluorescence in hiPSC-CMs with cytokine treatments.

**Figure 1—video 1.** Representative video of human induced pluripotent stem cell-derived cardiomyocytes (hiPSC-CM) contracting spontaneously in a culture dish.

https://elifesciences.org/articles/95867/figures#fig1video1

membrane potential (MMP) at baseline. For these experiments, cells were treated with cytokines at indicated concentrations for 48 hr. These studies revealed that each of the cytokines studied in our system did not alter MMP in hiPSC-CMs (*Figure 1—figure supplement 1C*). For the remaining studies, we focused our experiments on two of these cytokines that often are chronically increased in PLWH, TNF-α, and IFN-γ.

We next assessed whether treatment with TNF-α or IFN-γ alters diastolic function in hiPSC-CMs by measuring calcium transient and assessing the decay time and downstroke time in these cells using Single Cell Kinetic Image Cytometry (Vala Sciences). Treatment with both TNF-α and IFN-γ resulted in a significant increase in decay time and downstroke time, which are markers of cardiomyocyte relaxation, while beats per minute (indirect measurement of heart rate and not diastolic function) was essentially unchanged (*Figure 1A–C*). Additionally, treatment with mito-Tempo (an agent with mitochondrial antioxidant activity) reversed the changes that occurred with TNF-α (*Figure 1A and B*), indicating that a reduction in mitochondrial reactive oxygen species (ROS) may, at least partially, reverse the cardiomyocyte relaxation defect induced by TNF-α.

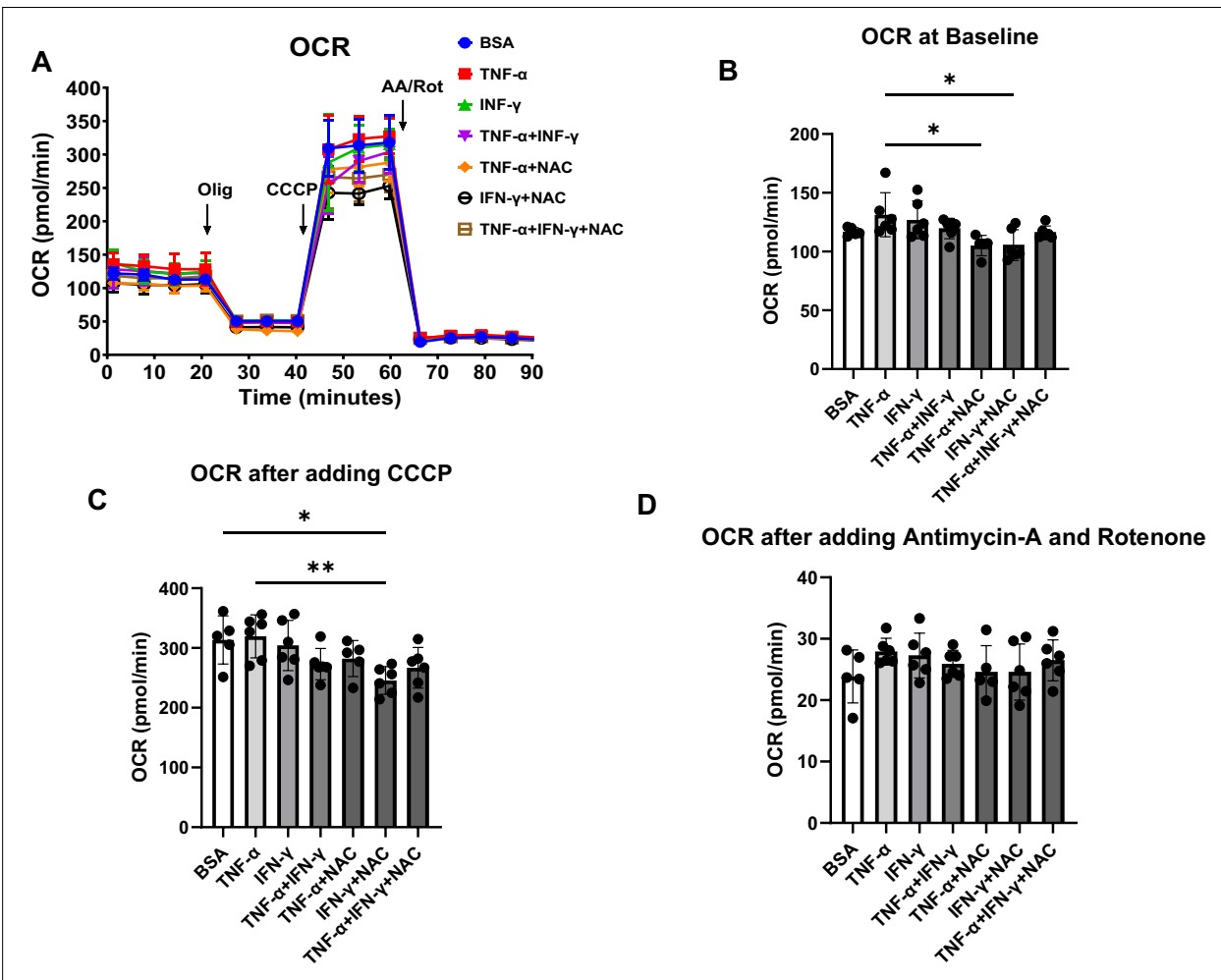

**Figure 2.** Oxygen consumption rate (OCR) in human induced pluripotent stem cell-derived cardiomyocytes (hiPSC-CM) treated with cytokines and antioxidant agent N-acetylcysteine (NAC). (**A**) OCR trace of hiPSC-CM treated with bovine serum albumin (BSA), TNF-α, IFN-γ, TNF-α + IFN-γ, TNF-α + NAC, IFN-γ + NAC, and TNF-α + IFN-γ + NAC. (**B–D**) Bar graph summary of data in (**A**) with OCR at baseline (**B**), after adding carbonyl cyanide m-chlorophenyl hydrazone (CCCP) (**C**), and after adding antimycin A and rotenone (**D**). N = 5–6. Data were analyzed by ordinary one-way ANOVA and post hoc Tukey's multiple-comparison test. Bars represent group mean, and error bars indicate standard error of the mean.

The online version of this article includes the following source data for figure 2:

**Source data 1.** OCR curve in hiPSCs treated with cytokines and NAC.

**Source data 2.** OCR at baseline, after CCCP treatment and after rotenone and antimycin A treatment.

Since mito-Tempo partially reversed the effects of TNF-α on calcium transient of hiPSC-CMs, we next assessed whether TNF-α and IFN-γ have any effects on mitochondrial respiration and whether these potential defects would be reversed with antioxidants. For these experiments, we used Seahorse XF96 Analyzer to measure the oxygen consumption rate (OCR) after treatment with TNF-α or IFN-γ in the presence and absence of the general antioxidant, N-acetylcysteine (NAC). While both TNF-α and IFN-γ did not induce significant changes in OCR, administration of NAC to hiPSC-CMs supplemented with TNF-α significantly reduced OCR at baseline (*Figure 2*). These data indicate that the beneficial effects of antioxidant reagents on calcium transient might be associated with the alleviation of aberrantly regulated OCR in TNF-α-treated cells. We next used this platform to study the effects of various drugs on in vitro measures of DD.

## Certain antiviral drugs and SGLT2 inhibitors reverse impaired relaxation in hiPSC-CMs

Given that we were able to induce impaired relaxation indicated by aberrant $Ca^{2+}$ uptake in hiPSC-CMs after treatment with inflammatory cytokines, we assessed whether various antiretrovirals and other drugs that potentially target the pathogenesis of HFpEF can reverse the dysregulation of calcium transient in vitro. We first assessed whether ART drugs at various concentrations cause cell death or increased ROS production in hiPSC-CMs. We treated hiPSC-CMs with tenofovir (a nucleotide-analog reverse transcriptase inhibitor), darunavir (a protease inhibitor), raltegravir, and elvitegravir (integrase inhibitors) at concentrations between 3 μM and 10 mM. The dose range was chosen to extend to tenfold above the IC50 concentrations and reflects the upper range of circulating drug concentration in patients receiving these medications (*Droste et al., 2005*; *Kakuda et al., 2014*; *Shiomi et al., 2015*; *Wang et al., 2011*). In this dose range, we did not observe apparent changes in cell viability and cellular ROS levels (*Figure 3—figure supplement 1A and B*), indicating that exposure to these single drugs alone has minimal effects on cell survival.

We then assessed whether ART can reverse the impaired relaxation induced by TNF-α. For these studies, we used the following single drug concentrations: 5 μM tenofovir, 10 μM darunavir, 3 μM raltegravir, and 10 μM emtricitabine (an HIV nucleoside analog reverse transcriptase inhibitor). As shown in *Figure 3*, while TNF-α increased the decay time in hiPSC-CMs, treatment with tenofovir, darunavir, raltegravir, and emtricitabine reversed these effects. These results indicate that ART may

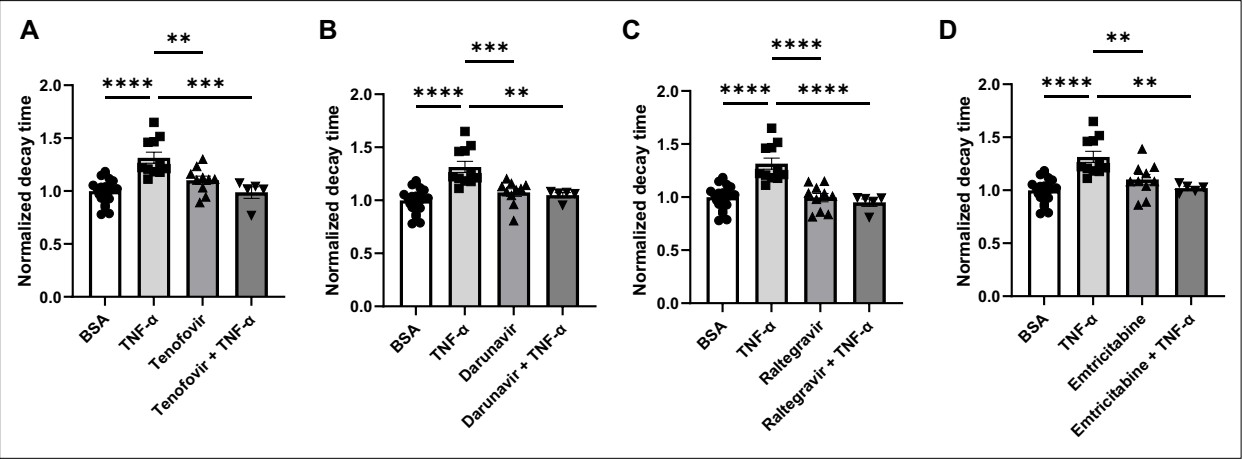

**Figure 3.** Decay time in human induced pluripotent stem cell-derived cardiomyocytes (hiPSC-CM) after treatment with TNF-α and various antiretroviral therapy (ART) drugs. (**A**) Decay time after treatment with TNF-α, tenofovir, and TNF-α+tenofovir. (**B**) Decay time after treatment with TNF-α, darunavir, and TNF-α+darunavir. (**C**) Decay time after treatment with TNF-α, raltegravir, and TNF-α + raltegravir. (**D**) Decay time after treatment with TNF-α, emtricitabine, and TNF-α + emtricitabine. N = 5–21. Data were analyzed by ordinary one-way ANOVA and post hoc Tukey's multiple-comparison test. Bars represent group mean, and error bars indicate standard error of the mean.

The online version of this article includes the following source data and figure supplement(s) for figure 3:

**Source data 1.** Normalized decay times in hiPSC-CMs treated with TNF-α and/or anti-HIV drugs.

**Figure supplement 1.** Cell viability and ROS levels in hiPSC-CMs treated with anti-HIV drugs.

**Figure supplement 1—source data 1.** Cell viability and ROS levels in hiPSC-CMs treated with anti-HIV drugs.

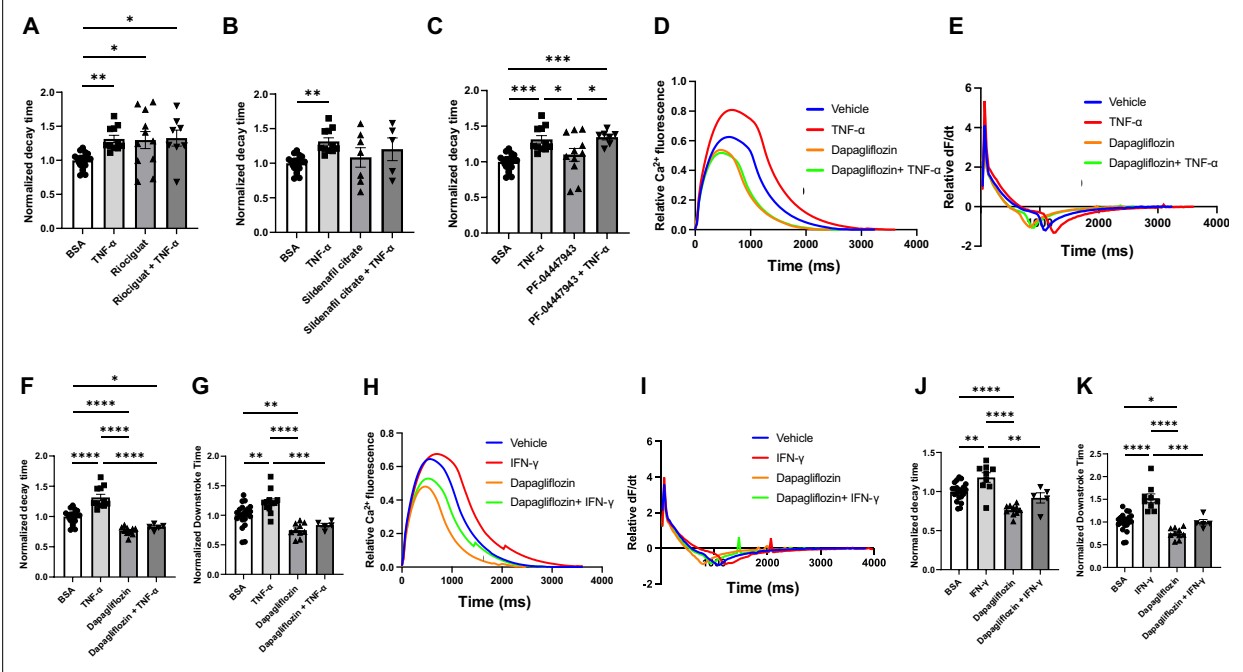

**Figure 4.** Decay time in human induced pluripotent stem cell-derived cardiomyocytes (hiPSC-CM) after treatment with TNF-α and riociguat, sildenafil citrate, PF-04447943, and dapagliflozin. (**A**) Decay time after treatment with TNF-α, sGS agonist riociguat, and TNF-α + riociguat (N = 8–21). (**B**) Decay time after treatment with TNF-α, sildenafil citrate, and TNF-α + sildenafil citrate (N = 5–21). (**C**) Decay time after treatment with TNF-α, PF-04447943, and TNF-α+PF-04447943 (N = 7–21). (**D**) Representative curve of calcium transient in hiPSC-CM treated with TNF-α, dapagliflozin, and TNF-α + dapagliflozin. (**E**) Curve of derivative calculated with data in (**D**). (**F**) Decay time after treatment with TNF-α, dapagliflozin, and TNF-α+dapagliflozin (N = 5–21). (**G**) Downstroke time after treatment with TNF-α, dapagliflozin, and TNF-α + dapagliflozin(N = 5–21). (**H**) Representative curve of calcium transient in hiPSC-CM treated with IFN-γ, dapagliflozin, and IFN-γ + dapagliflozin. (**I**) Curve of derivative calculated with data in (**H**). (**J**) Decay time after treatment with INF-γ, dapagliflozin, and IFN-γ + dapagliflozin (N = 5–21). (**K**) Downstroke time after treatment with INF-γ, dapagliflozin, and IFN-γ + dapagliflozin (N = 5–21). Data were analyzed by ordinary one-way ANOVA and post hoc Tukey's multiple-comparison test. Bars represent group mean, and error bars indicate standard error of the mean.

The online version of this article includes the following source data and figure supplement(s) for figure 4:

**Source data 1.** Normalizaed decay times in hiPSC-CMs treated with TNFα and Riociguat, Sildenafil citrate and PF-04447943.

**Source data 2.** Analyses of calcium transient in hiPSC-CMs treated with TNFα, INFγ and 10 μM Dapagliflozin.

**Figure supplement 1.** Analyses of calcium transient in hiPSC-CMs treated with TNFa and/or 1 μM Dapagliflozin.

have beneficial effects on the inflammation-mediated deterioration of calcium transient that occurs in HIV patients, raising the possibility that its potential benefits may go beyond their inhibitory effects on the virus itself.

A number of other drugs have also been proposed to potentially exert beneficial effects in HFpEF. Among these, we focused on an oral soluble guanylate cyclase stimulator (riociguat,1 μM), a phosphodiesterase (PDE)-5 inhibitor (sildenafil, 1 μM), a PDE9 inhibitors (PF-04447943, 5 μM), and an SGLT2 inhibitor (dapagliflozin, 10 μM). The dosages are based on previous functional studies in isolated cardiomyocytes (*Castro et al., 2006*; *Kiso et al., 2013*; *Lee et al., 2015*; *Reinke et al., 2015*; *Zhang et al., 2006*). Specifically, we also tested 1 μM of dapagliflozin and found no effects on decay and downstroke times (*Figure 4—figure supplement 1A and B*). While TNF-α caused a significant increase in decay time in hiPSC-CMs, riociguat, sildenafil, and PF-04447943 failed to reverse this effect (*Figure 4A–C*), indicating that they do not have the properties to reverse the inflammation-mediated impaired relaxation. However, SGLT2 inhibitor dapagliflozin significantly reversed the prolonged decay and downstroke times induced by TNF-α in hiPSC-CMs (*Figure 4D–G*). Additionally, we repeated the experiments with IFN-γ and dapagliflozin and showed that the drug was able to reverse the increased decay and downstroke time associated with IFN-γ as well (*Figure 4H–K*). These studies are in accordance with the clinical studies that have shown beneficial effects of SGLT2

**Table 1.** Clinical characteristics of study population for group 1.

| Participant | Age | High or low myocardial perfusion reserve group | Sex | CD4 count | Diabetes mellitus? | Active cigarette smoking? | Hypertension? | Coronary artery disease diagnosis? | Left ventricular ejection fraction | Myocardial perfusion reserve |
|---|---|---|---|---|---|---|---|---|---|---|
| 1 | 50 | High | Male | 487 | No | Yes | Yes | No | 64 | 2.74 |
| 2 | 62 | Low | Male | 703 | No | No | Yes | Yes | 59 | 1.03 |
| 3 | 64 | High | Male | 321 | No | No | No | Yes | 65 | 3.43 |
| 4 | 55 | High | Male | 749 | No | No | Yes | Yes | 59 | 1.90 |
| 5 | 71 | Low | Male | 1743 | No | No | Yes | Yes | 55 | 1.03 |
| 6 | 59 | Low | Male | 678 | No | No | Yes | No | 52 | 1.13 |

inhibitors in HFpEF and suggest that they may also benefit patients with inflammation-induced aberrant calcium transient, like HIV+ patients.

## Treatment with serum from patients with HIV does not induce impaired relaxation in hiPSC-CMs

Given that we could simulate impaired calcium transient in hiPSC-CMs using inflammatory cytokines, we next assessed whether short-term treatment of these cells with serum from HIV+ patients with evidence of DD would also induce impaired $Ca^{2+}$ uptake in the cells. We used two sources of serum in our studies. For the first group, we recruited six men with chronic HIV on ART with undetectable viral loads who underwent cardiac magnetic resonance imaging (CMRI) at the Northwestern Memorial Hospital (NMH) for the assessment of myocardial perfusion reserve (MPR). Clinical and MRI-based characteristics of these patients are included in *Table 1*. Lower values of MPR indicate impaired global myocardial blood flow in response to vasodilator stress and are associated with higher incidence of HFpEF and poor prognosis (*Arnold et al., 2022*; *Markley et al., 2023*). The second group included well-controlled patients with HIV at University of California San Francisco (UCSF) with echocardiographic evidence of normal cardiac function or DD.

For both groups, we treated hiPSC-CMs with 10% serum from these patients for 48 hr, followed by the measurement of calcium transients. Additionally, we treated human umbilical vein endothelial cells (HUVEC) with the serum of these patients for 20 hr and assessed their angiogenic potential given the critical relationship between endothelial dysfunction and the development of HFpEF (*Cornuault et al., 2022*; *Paulus and Tschöpe, 2013*). For the first group of patients, we stratify into higher (1.9–3.4; N = 3) versus lower (1.0–1.1; N = 3) MPR values, which correspond to control and DD, respectively. As shown in *Figure 5A–C*, serum from HIV+ patients with DD from group 1 did not result in a significant change in decay and downstroke time in hiPSC-CMs. Similar results were obtained after 24 hr incubation. Additionally, serum from these patients did not alter the tube network formation of HUVECs (*Figure 5D–G*, *Figure 5—figure supplements 1 and 2*). We also assessed the effects of serum from HIV+ patients from group 2. The control group for experiments included serum from HIV patients without evidence of DD. Our data indicate that the treatment of hiPSC-CMs and HUVECs with 2% serum of these patients for 3 or 24 hr does not alter decay time (*Figure 6A–D*) and tube formation capacity (*Figure 6E–I*), respectively. For hiPSC-CMs studies, we exposed the cells to serum for 3 hr and 24 hr and did not observe a difference in decay time with either time point. These results suggest that short treatment of hiPSC-CMs or HUVECs with serum from patients with DD does not induce changes in calcium transient or their angiogenic potential, respectively.

## Discussion

HFpEF is a common disease and its incidence continues to increase. Among the risk factors for HFpEF, inflammation is believed to be a common mechanism for the development of the disease. The chronic inflammatory condition associated with treated HIV infection is believed to be responsible for the development of HFpEF in these patients. Despite the high incidence of this disorder among the general population and those with HIV, there are currently very few systems available to study this

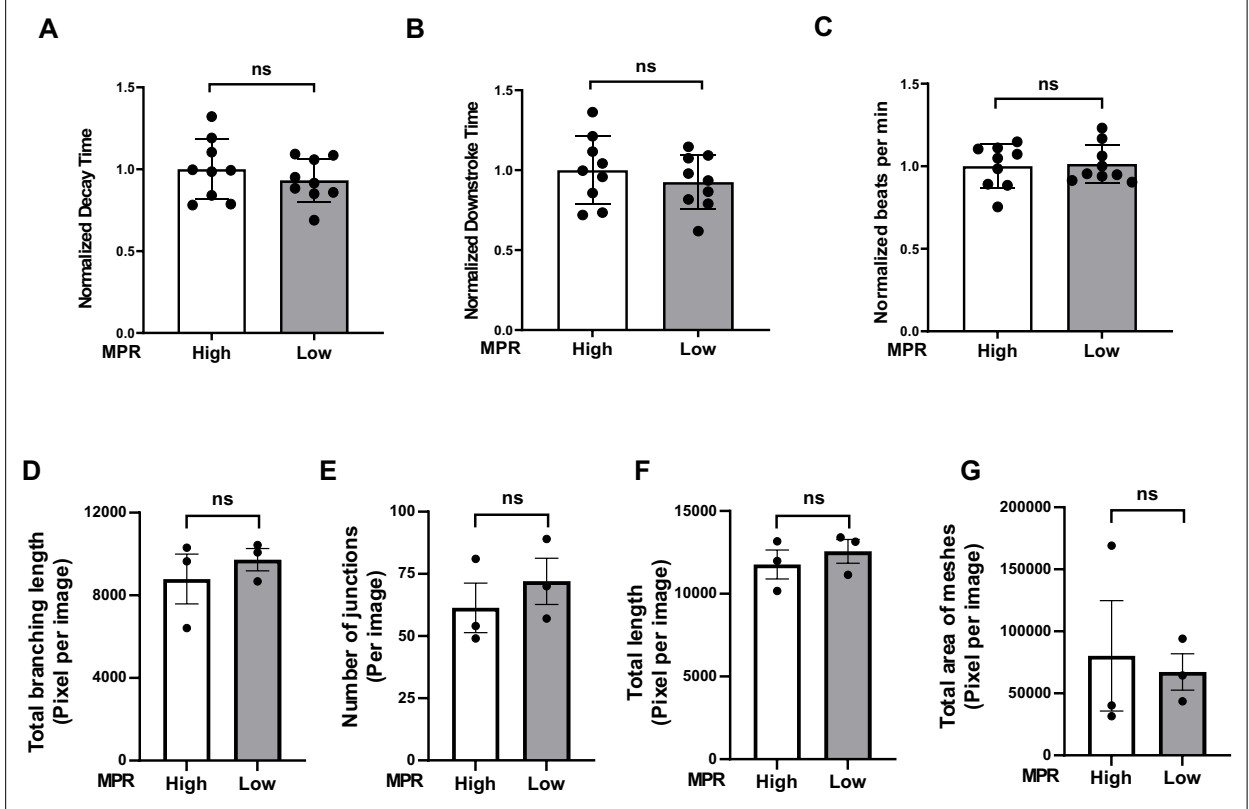

**Figure 5.** Ca²⁺-decay time of human induced pluripotent stem cell-derived cardiomyocytes (hiPSC-CM) and angiogenic function of ECs after treatment with serum from group 1 HIV+ patients with diastolic dysfunction (DD). (**A–C**) Pooled normalized decay time (**A**), downstroke time (**B**), and beating rate (**C**) in hiPSC treated with 10% serum for 48 hr from group 1 HIV+ patients with DD (3 values from cells treated with each serum). (**D–G**) Angiogenic parameters, including total length of branching (**D**) (N = 3), number of junctions (**E**) (N = 3), total length (**F**) (N = 3), and number of meshes (**G**) (N = 3) in human umbilical vein endothelial cells (HUVEC) after treatment with 2% serum from group 1 patients for 20 hr. Data were analyzed by unpaired Student's *t*-test. Bars represent group mean, and error bars indicate standard error of the mean.

The online version of this article includes the following source data and figure supplement(s) for figure 5:

**Source data 1.** Analyses of calcium transient in hiPSCs treated with serum from HIV patients.

**Source data 2.** Quantitative analyses of tube formation assay in HUVECs treated with serum from HIV patients.

**Figure supplement 1.** Representative images of formed tube structure with human umbilical vein endothelial cells (HUVEC) treated with 2% serum from HIV+ patients with high or low myocardial perfusion reserve (MPR) for 20 hr.

**Figure supplement 2.** Representative magnified images of formed tube structure with human umbilical vein endothelial cells (HUVEC) treated with 2% serum from HIV+ patients with high or low myocardial perfusion reserve (MPR) for 20 hr.

disorder and screen for drugs that can target the disease. In this article, we used hiPSC-CMs as a platform to assess whether abnormal cardiomyocyte relaxation can be induced in these cells in response to inflammatory cytokines. We demonstrate that TNF-α and IFN-γ can induce prolonged decay time in hiPSC-CMs, which can partially be reversed with mitochondrially targeted antioxidants. We also demonstrate that most ART drugs and SGLT2 inhibitors can reverse the aberrant calcium transient associated with inflammatory cytokines. Finally, to determine whether treatment of hiPSC-CMs serum from patients with HIV and DD can also induce deterioration of calcium transient in hiPSC-CMs, we treated these cells with patient serum, but were not able to mimic DD. We also assessed whether serum from these patients can induce defects in angiogenic potential and found no effects on these parameters. Overall, our results provide a novel platform to study dysregulated calcium transient associated with inflammatory cytokine treatment and test the effectiveness of various agents in this disorder.

Studying HFpEF in tissue culture has its own limitations since HFpEF is a systemic disorder and usually develops as a result of a number of risk factors, including diabetes, hypertension, and chronic kidney disease. It is important to note that DD is a major component of HFpEF and studying the

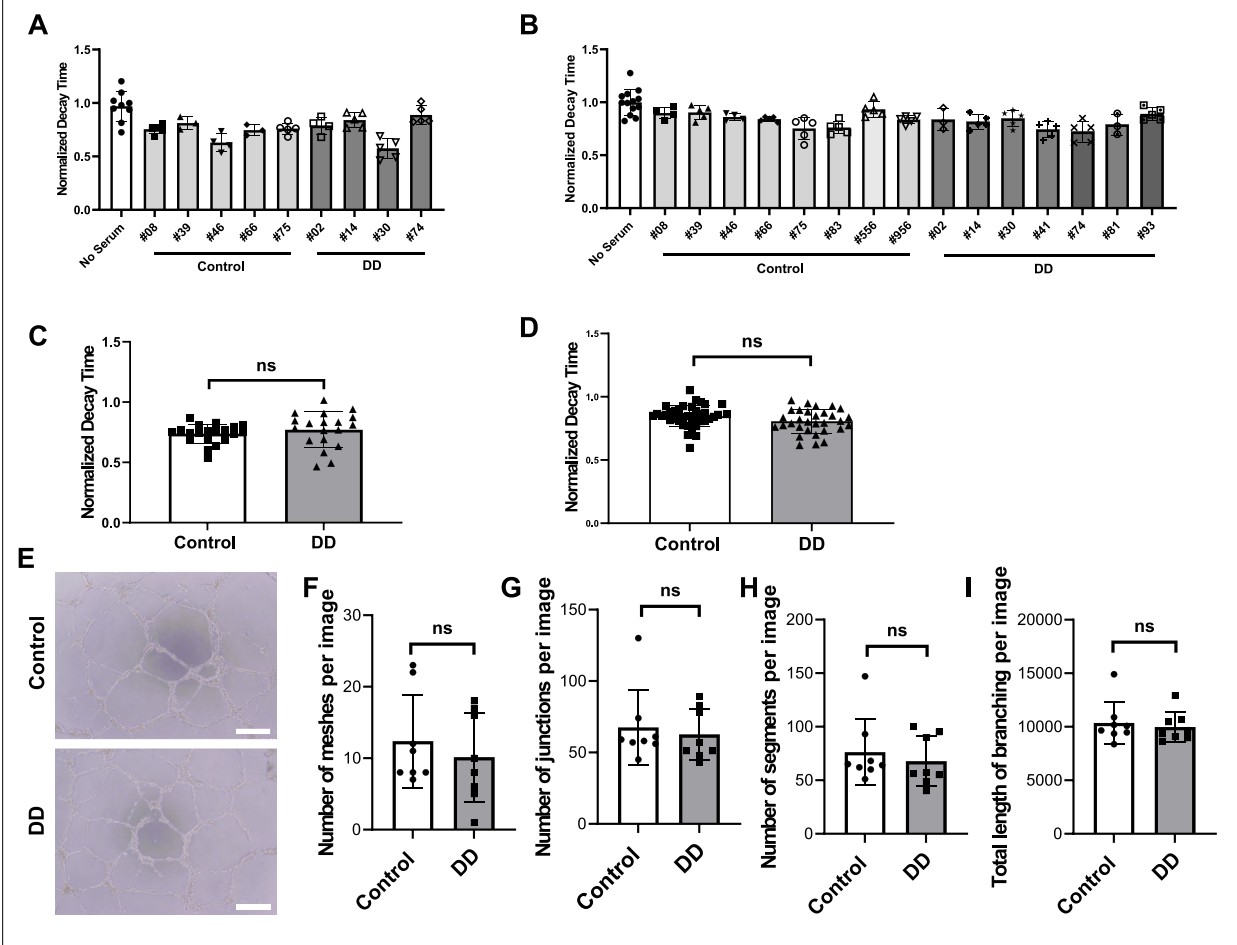

**Figure 6.** Diastolic function of human induced pluripotent stem cell-derived cardiomyocytes (hiPSC-CM) and angiogenic function of ECs after treatment with serum from HIV+ patients with diastolic dysfunction (DD) (group 2). (**A, B**) Individual normalized decay time 3 hr (N = 3–9) (**A**) or 24 hr (N = 3–13) (**B**) after treatment of hiPSC-CM with 2% serum from patients. (**C, D**) Pooled decay time data from patients in (**A**) and (**B**) with 3 hr (**C**) and 24 hr (**D**) after treatment with serum (N = 21–22 for (**C**) and N = 31-38 for (**D**)). (**E**) Representative image of formed tube structure with cultured human umbilical vein endothelial cells (HUVEC) treated with serum from HIV+ patients with DD. Scale bars indicate 250 μm. (**F–I**) Assessment of tube formation of HUVECs 20 hr after treatment with serum of patients, including number of meshes (**F**) (N = 8), number of junctions (**G**) (N = 8), number of segments (**H**) (N = 8), and total length of branching (**I**) (N = 8). Data were analyzed by unpaired Student's *t*-test for (**C, D, F, G, H, I**). Bars represent group mean, and error bars indicate standard error of the mean.

The online version of this article includes the following source data for figure 6:

**Source data 1.** Normalized decay time in iPS-CMs treated with HIV patient serum with or without diastolic dysfunction.

**Source data 2.** Quantitative analyses of tube formation assay in HUVECs treated with serum from HIV patients.

relaxation abnormalities associated with this disorder, especially in tissue culture setting, can provide major clues to this disease at the cellular level. Thus, using a cell culture system to mimic the functional abnormality noted in this disorder can provide clues to the disease that can later be tested at the organismal level. Given that there are limited animal models for HFpEF, an in vitro model will provide a powerful platform for hypothesis generation and drug screening prior to committing resources for studying these findings at the organismal level.

We demonstrated that treatment with certain cytokines can induce prolonged decay time in hiPSC-CMs, but serum from HIV patients with imaging evidence of DD did not cause similar abnormalities. This could be due to several reasons, including short duration of the treatment with human serum, lower concentration of cytokines in the serum compared to the concentration we used in our studies, and possibly differences in gene expression patterns between human matured cardiomyocytes and hiPSC-CMs. In fact, previous studies demonstrated lower concentrations of TNF-α and IFN-γ (41.32 pg/ml and 0.78 pg/ml, respectively) in HIV+ patient serum than ones adopted in this

study (*Musa et al., 2021*; *Osuji et al., 2018*). Because of the limited availability of serum samples from HIV patients, we were not able to test different concentration and duration of serum treatment in our hiPSC-CM system. Further experiments with serum from HIV+ with DD in a large scale are warranted to determine whether other conditions can induce aberrant calcium transient. Nevertheless, it is important to note that cytokines alone can induce impaired $Ca^{2+}$ uptake in hiPSC-CMs and can be used to induce this condition in vitro.

It is surprising that SGLT2 inhibitors reversed the progression of DD in our system since cardiomyocytes do not express SGLT2 under physiological conditions. Interestingly, Marfella et al. reported that adult cardiomyocytes express SGLT2 in diabetic conditions (*Marfella et al., 2022*). Since diabetes is associated with low-grade systemic inflammation, HIV patients might also express SGLT2 in their cardiomyocytes. Taken together, our data indicate that SGLT2 inhibitors at the dosages used in this study reverse the impaired calcium transients induced by TNF-α and IFN-γ. Additionally, the fact that ART drugs were able to reverse the impaired $Ca^{2+}$ uptake associated with cytokine treatment raises the possibility that these drugs can prevent damage to cardiomyocyte function as a result of the inflammation associated with HIV, in addition to their anti-retroviral effects.

There are limitations to our studies. Although decay and downstroke time were utilized as surrogates of cardiomyocyte relaxation, direct biomechanical characterization of contraction was not conducted in our study. While cytosolic calcium concentration is a major factor in the regulation of cellular relaxation (*Feridooni et al., 2015*), certain conditions, including the transition of sarcomere protein isoforms to pathogenic ones (*Fentzke et al., 1999*) and stimulation of β-adrenergic receptors (*Bers et al., 2019*), can alter this relationship. However, since hiPSC-CMs utilized for each study are from iPS cells derived from a single donor, we believe that these parameters had minimal effects on our system. We also did not elucidate the molecular mechanisms underlying prolonged decay time induced by TNF-α and IFN-γ. TNF-α treatment is reported to induce a longer decay of calcium transient and a decreased sarcoplasmic ATPase (SERCA) expression in rabbit cardiomyocytes from pulmonary vein (*Lee et al., 2007*) Thus, our observation in hiPSC-CMs may also be through decreased expression of SERCA though further studies remain conducted. Darunavir is generally used in combination with ritonavir and elvitegravir along with cobicistat in clinical practice. These additional drugs were not tested in our studies. Additionally, our studies employed only single drugs, which is not commonly used in the clinic. Also. we did not perform a dose–response curve with the drugs we tested, but used concentrations that were reported previously. Another limitation of our study is that we did not determine the mechanism of how ART drugs with such different effects on HIV proteins can have an effect on cellular relaxation. This is an important issue, and further studies are needed to better assess the mechanism of how ART can exert such effects.

In summary, we provide a novel approach to study inflammation-induced calcium transient at the cellular level using hiPSC-CMs. We use this method to test a number of drugs and whether they can reverse the prolonged decay time associated with inflammation. This system can prove useful in studying the molecular basis and potential treatments for HFpEF induced by chronic inflammation.

## Materials and methods
### Human induced pluripotent cell derivation and cardiac differentiation
Protocols and consents were approved by the institutional review board of Northwestern University (STU00204341). The hiPSC from healthy individuals was generated following the protocol published before (*Churko et al., 2013*). Identity of hiPSC was authenticated by the confirmation of karyotype, gene expression, and undifferentiated cell marker expression (*Lyra-Leite et al., 2023*). All cultures were routinely tested for mycoplasma using a MycoAlert PLUS Kit (Lonza, LT07-710) and a Varioskan LUX (Thermo) plate reader. The hiPSC were maintained and differentiated into beating cardiomyocytes, according to the previously published protocol (*Burridge et al., 2015*).

### Cell line
HUVECs were purchased from Lonza (#C2519A) and subsequently used in our experiments. The cell identity was authenticated by the vender, confirming double-positive expression for CD31 and CD105. According to the certificate of analyses provided by the vender, the cells tested negative for mycoplasma, and all cultures were routinely tested for mycoplasma as well.

## MMP measurement

Media was removed from hiPSC-CM and were washed with phosphate-buffered saline (PBS) and FluoroBrite DMEM (no glutamine, no HEPES, no phenol red, no sodium pyruvate, Thermo A1896702) was added. Cells were stained with 5 nM TMRE (for mitochondrial stain) for 20 min and then washed with PBS before immunofluorescence. TMRE (red channel) signal was analyzed as MMP using ImageJ (Fiji).

## Seahorse assay

7 days before assay, hiPSC-CMs were plated at 100,000 cells per well in Matrigel-coated Seahorse XF cell culture 96-well plate. The day before the assay, the Seahorse cartridge was placed in the XF calibrant and incubated overnight at 37°C. On the day of the assay, the plates were incubated at room temperature for 1 hr in glucose-free complete DMEM or RPMI without bicarbonate or phenol red to allow even distribution of cells across the well floor. Before placing the sample plates in the Seahorse XF96 Analyzer, media volume was adjusted to 150 µl in each well. 25 mM glucose, 1.5 mM oligomycin, 1 µM CCCP, and 20 µM rotenone (Rot)/antimycin A (AA) were diluted in DMEM and injected sequentially into each well, following the standard Seahorse protocol. Cytokine treatment was started 48 hr before Seahorse assay.

## Treatment of hiPSC with IFN-γ and TNF-α

hiPSC-CMs after 21–24 days culturing were seeded into 6-well or 96-well cell culture plates. After 5 days, IFN-γ at 1 ng/ml and TNF-α at 300 pg/ml were added to the cells for 48 hr.

## Drug concentrations

We used the following drug concentrations for our studies: riociguat 1 µM, sildenafil citrate 1 µM, PF-04447943 5 µM, dapagliflozin 1 or 10 µM, tenofovir 5 µM, emtricitabine 10 µM, darunavir 10 µM, and raltegravir 3 µM.

## Calcium transient studies in hiPSC-CM

For $Ca^{2+}$ transient analysis, day 21–24 hiPSC-CMs were plated on the Matrigel-coated 96-well black cell culture plate at a density of 75,000 cells/well. Calcium imaging was performed as described previously (*Romero-Tejeda et al., 2023*). On the day of imaging (days 28–32), cells were stained with a buffer containing 5 µM Fluo-4AM, NucBlue (2 drops/10 ml), 2.5 mM probenecid, 0.02% Pluronic F-127 in FluoroBrite DMEM for 1 hr at 37°C, 5% $CO_2$. After 1 hr incubation, aspirate dye loading solution and add 200 µl FluoroBrite DMEM to each well for imaging. Single Cell Kinetic Image Cytometry (Vala Sciences) was used to scan the plate and CyteSeer Automated Video Analysis Software (Vala Sciences) was used to analyze the $Ca^{2+}$ transient.

## Tube formation assay

The formation of tube networks was assessed as described before (*Das et al., 2018*). HUVECs were seeded at 20,000 per well in a 96-well plate coated with 75 ml Matrigel (Fisher Scientific) or Cultrex (R&D Systems) reduced growth factor basement membrane matrix. The cells were treated with EBM-2 medium containing 2% serum from HIV+ patients obtained from NMH and UCSF wherever mentioned. Following a 20 hr incubation, tube networks from each biological replicate were analyzed in at least three random fields by light microscopy. The number of branch points (junctions), segments and meshes, and total length of tubule networks (total length of branching) was quantified by Fiji software (Angiogenesis Analyzer).

## Participant samples

Blood samples from six men with chronic HIV on ART with undetectable viral loads who underwent CMRI for the assessment of MPR were used. For CMRI, all scanning was performed on 1.5 T MRI scanners (Siemens Medical Systems), with participants receiving FDA-approved intravenous double-dosing of gadolinium contrast followed by adenosine as the vasodilator agent. Images were performed at maximal hyperemia (vasodilation) and compared with rest: MPR was calculated as the ratio of stress myocardial blood flow to rest myocardial blood flow as described previously (*Lee and Johnson, 2009*). Comparisons in hiPSC-CM performance characteristics were made between those with higher (1.9–3.4; N = 3) versus lower (1.0–1.1; N = 3) MPR values; additional clinical and MRI-based

characteristics are included in *Table 1*. The study was approved by the Institutional Review Board of Northwestern University (STU00204874).

Serum samples from UCSF are from an IRB-approved (IRB# 10-03112) longitudinal cohort study evaluating the pathogenesis of HIV and HHV-8 to pulmonary hypertension. PLWH underwent serial phlebotomy, transthoracic echocardiogram, flow-mediated dilation, 6 min walk test, and pulse wave velocity measurements. A subset of participants underwent a right heart catheterization. The study is IRB-approved, and participants provided signed informed consent prior to undergoing study procedures. Serum samples are derived from documented HIV-positive individuals on stable ART and have diastolic function assessed by echocardiogram. Samples are de-identified and stored long term with UCSF's AIDS Specimen Bank (ASB).

Of the 16 participants included from UCSF, median age was 54.1 (IQR 45.0, 57.1), median duration of HIV infection was 17 years (IQR 12.7, 22.7), median CD4 count was 395 (IQR 334, 645), 88% were male, 50% were Caucasian, 37% were African American, and 13% were Hispanic/Latino. Comorbidities included 43.8% had hypertension, 31.3% had hyperlipidemia, 12.5% had diabetes, and 62.5% were current or past smokers. All specimens were matched case–control to those with DD to those without DD.

Baseline echocardiogram characteristics included everyone with preserved LV systolic function (≥50%). In the DD group, n = 2 had grade 1 DD, n = 3 had grade 2 DD, and n = 3 had grade 3 DD. Two individuals in the DD group had mild left atrial enlargement (LA volume index 35–41 ml/m$^2$). Five individuals in the DD group and two in the control group had RVSP ≥ 35 mmHg.

Twelve participants (six from each group) underwent right heart catheterization with two in the DD group having elevated mean pulmonary arterial pressure (≥25 mmHg). Three participants in the DD group and two in the control group had right atrial pressure >5 mmHg.

## Statistical analysis

Data are presented as mean ± SEM. Unpaired Student's *t*-test and one-way ANOVA, with post hoc Tukey's test were conducted to assess statistical significant difference among experimental groups. A p-value <0.05 was considered statistically significant.

## Acknowledgements

Funding for this work was provided through NHLBI HL140973 awarded to HA and the American Heart Association grant SP0065527 to YT. HA and YT conceptualized the project and wrote the paper. CC, JS, H-CC, MB, and DL-L, carried out the experiments. MF and PH provided human samples. PWB and RD provided scientific guidance.

## Additional information

### Competing interests

Hossein Ardehali: Reviewing editor, *eLife*. The other authors declare that no competing interests exist.

### Funding

| Funder | Grant reference number | Author |
| --- | --- | --- |
| National Heart, Lung, and Blood Institute | R01 HL140973 | Hossein Ardehali |
| American Heart Association | SP0065527 | Yuki Tatekoshi |

The funders had no role in study design, data collection and interpretation, or the decision to submit the work for publication.

### Author contributions

Yuki Tatekoshi, Conceptualization, Formal analysis, Investigation, Visualization, Writing - original draft, Writing - review and editing; Chunlei Chen, Software, Investigation, Visualization, Methodology,

Writing - original draft, Project administration, Writing - review and editing; Jason Solomon Shapiro, Hsiang-Chun Chang, Conceptualization, Formal analysis, Supervision, Investigation; Malorie Blancard, Resources, Software, Formal analysis, Investigation; Davi M Lyra-Leite, Resources, Software, Investigation; Paul W Burridge, Resources, Software, Supervision, Validation; Matthew Feinstein, Priscilla Hsue, Resources; Richard D'Aquila, Resources, Supervision; Hossein Ardehali, Conceptualization, Resources, Supervision, Funding acquisition, Validation, Writing - original draft, Project administration, Writing - review and editing

### Author ORCIDs
Yuki Tatekoshi ⓘ http://orcid.org/0000-0001-7345-0158
Jason Solomon Shapiro ⓘ https://orcid.org/0000-0003-0880-3142
Davi M Lyra-Leite ⓘ https://orcid.org/0000-0002-2929-5100
Richard D'Aquila ⓘ https://orcid.org/0000-0002-9653-5987
Hossein Ardehali ⓘ https://orcid.org/0000-0002-7662-0551

### Ethics
Protocols and consents for the generation of hiPSCs and the differentiation into hiPSC-CMs were approved by the institutional review board (IRB) of Northwestern University (STU00204341). Cardiac MRI study and the serum collection from HIV patients were approved by the IRB of Northwestern University as well (STU00204874). Participants in UCSF whose serum and clinical physiological data were used for this study provided signed informed consent prior to undergoing study procedures, and the study is an IRB-approved cohort study in UCSF (IRB# 10-03112) .

Reviewer #1 (Public review): https://doi.org/10.7554/eLife.95867.3.sa1
Reviewer #2 (Public review): https://doi.org/10.7554/eLife.95867.3.sa2
Author response https://doi.org/10.7554/eLife.95867.3.sa3

## Additional files

### Supplementary files
• MDAR checklist

### Data availability
All data generated or analyzed during this study are included in the manuscript and supporting files; source data files have been provided for Figures 1-6, Figure 1—figure supplement 1, Figure 3—figure supplement 1 and Figure 4—figure supplement 1.

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
