## [Editor Report · eLife assessment]

This **useful** study focuses on heart failure with preserved ejection fraction (HFpFE), common in patients with HIV. Researchers used induced human pluripotent stem cell-derived cardiomyocytes (hiPSC-CMs) to stimulate (HEFpEF) and found that inflammatory cytokines alter Ca2+ transients. SGLT2 inhibitors and mitochondrial antioxidants reversed this effect. While the study is **incomplete** and preliminary, its strength lies in introducing hiPSC-CMs as a tool for investigating HFpEF mechanisms. A major weakness was found to be limited functional assessment relevant to HFpEF.

---

## [Referee Report · Reviewer #1 (Public review)]

Summary:

This is a reviewed manuscript submission to better understand mechanisms for why HIV individuals have diastolic dysfunction. Due to a lack of robust animal models, the team developed iPS-CM models to study HFpEF. The revised manuscript has toned down claims regarding diastolic function given the lack of mechanical testing. The team has focused on the altered Ca2+ phenotype, which improves the precision of the claims of the team. There remain questions on the functional relevance of the altered calcium handling given the lack of physiological assays. There also remain some questions about whether SGLT2 protein is expressed in these models without testing it, and whether the effects of SGLT2i could be off-target.

Overall, the revised manuscript is improved. I have no major remaining concerns except that the lack of biomechanical assessments diminishes the significance of the study as altered calcium alone would not be considered sufficient evidence for diastolic dysfunction, which was major task set out to answer by the group.

---

## [Referee Report · Reviewer #2 (Public review)]

The authors investigated the role of inflammatory molecules in diastolic dysfunction and screened antiviral and cardioprotective pharmacological agents for their potential to reverse inflammation-mediated diastolic dysfunction. This study focuses on heart failure with preserved ejection fraction (HFpEF) in people living with HIV (PLWH), a condition often challenging to study due to the lack of suitable animal models. Using human induced pluripotent stem cell-derived cardiomyocytes (hiPSC-CMs), researchers simulated HFpEF in vitro. They observed that inflammatory cytokines impaired cardiomyocyte relaxation, mimicking HFpEF, while SGLT2 inhibitors and mitochondrial antioxidants reversed this effect. Exposure to serum from HIV patients did not induce dysfunction in hiPSC-CMs. These findings suggest hiPSC-CMs as a promising model for understanding HFpEF mechanisms and testing potential treatments.

Comments on revised version:

The revised manuscript has been improved satisfactorily. The authors also have addressed all of my concerns.

---

## [Author Response]

The following is the authors’ response to the original reviews.

**Public Reviews:**

**Reviewer #1 (Public Review):**
I am not convinced how this study relates to HIV individual HFpEF, and the study design does not seem to be well thought out.

This is an important point and we have modified the manuscript as mentioned below in our responses.

The connectivity of the study experiments is loose, and data analysis and conclusions are broadly overstated and misinterpreted.

We have modified the manuscript thoroughly so the data are interpret properly, and the conclusions are stated logically.

For example the study lacks any measure of diastolic contractile function, and even if performed, the relevance of TNFa treatments to cells in vitro in these immature cell contexts would remain unclear. There is surprisingly no reported molecular analyses of potential mechanisms of the calcium transient changes. The study falls short in molecular detail and instead relies on drug treatments and responses that are hard to interpret with dosages that are not well justified and treatments that are numerous. Unclear what changes in calcium transients mean functionally without a comprehensive assessment of CM biomechanical contraction and relaxation measurements, and this would also require parallel molecular investigations of potential targets of any phenotypes observed.

As mentioned above, we have modified the manuscript so the data are interpret properly, and the conclusions are stated logically. In terms of mechanisms for the observed phenomenon, we agree that this was not the focus of studies, however, we have provided a paragraph in the discussion that covers this topic. Although Decay and downstroke time were utilized as surrogates of cardiomyocyte relaxation, direct biomechanical characterization of contraction was not conducted in this study. While cytosolic calcium concentration is a predominant factor to regulate the cell’s relaxation (Reference 52 in the manuscript), there are several mechanisms to modify the relationship, including the transition of sarcomere protein isoforms to pathogenic ones (Reference 53 in the manuscript) and the stimulation of β-adrenergic receptor on cardiomyocytes (Reference 54 in the manuscript). Since hiPSC-CMs utilized for each study is from iPS cells derived from a single donor, we believe that the patterns of sarcomere protein expression and the regulation of β-adrenergic receptor pathway should be consistent among samples, supporting their effects should be minimum in our system. We also did not elucidate molecular mechanisms underlying prolonged decay time induced by TNF-α and IFN-γ in this study. Lee et al. reported that 25 ng/ml TNFa treatment induced a longer decay portion of the calcium transient and a decreased sarcoplasmic ATPase (SERCA) expression in rabbit cardiomyocytes from pulmonary vein (Reference 55 in the manuscript), suggesting our observation in iPS-CM is also through decreased expression of SERCA though further studies remain conducted.

Calcium transient data need to be better illustrated such as with representative peak tracings. The data overall is with too few samples, particularly given the inherent heterogeneity of iPSCM studies. The iPS-CM system as a model for diastolic dysfunction remains unestablished.

We have now prepared several representative curves of calcium transient and their derivatives in Figure 4 D and E, H and I, and in Figure 1-figure supplement 1B. In terms of the way to collect Ca-transient data, each dot in the bar graphs represents the average of signals obtained from one well of the 96-well plates. About 75K cells were seeded in one well, and we believe that the number of cells integrated in the analyses should be sufficient for the statistical analyses. We modified our manuscript as this system does not quantifying diastolic function directly, but represents Ca measurements that indicate cardiomyocyte relaxation.

There are unclear dose choices for the various ART drugs tested, as well as the other drugs tested such as SGLT2i. Besides the observation that SLC5A2 (SGLT2 target) is not established to be expressed in adult mammalian cardiomyocytes.

Thank you for the comment. The dose ranges of ART drugs were chosen to extend to 10fold above the IC50 concentrations and reflects the upper range of circulating drug concentration in patients receiving these medications (Reference 36-39 in the manuscript). For SGLT2 inhibitor concentration, we referred to a paper utilizing 1-10 μM dapagliflozin (PMID: 35818731). We conducted a preliminary study to test the effect of 1 and 10 μM of dapagliflozin on the Ca-transient of iPS-CMs, and we found that 1 μM of the drug treatment did not cause changes in Ca-transient. Marfella et al. reported that SLC5A2 (SGLT2) expresses in cardiomyocytes under diabetic condition (PMID 36096423). Since diabetes is associated with low grade systemic inflammation, HIV patients might also express SGLT2 in cardiomyocytes. Taken together, we believe that the dosages of the drugs used in our studies are relevant to the clinical therapeutical usages of the drugs.

HIV plasma samples were not tested for cytokine levels, but this could be done to assess the validity of the final experiments. It is unclear what is being tested with these experiments.

This is a good point and we agree with the reviewer. However, we had limited amount of the patient serum and could not perform a comprehensive analysis of these samples. Nevertheless, we have added a section in the Discussion section providing some clinical relevance of our findings based on the papers that have assessed cytokine levels in the serum of HIV patients.

The choice of serum controls from a second institution (UCSF) opens up concerns over batch effects unrelated to differences in diastolic dysfunction. However, there were no differences with the Northwestern samples. It is unclear why this data is included as it does not add to the impact of the study.

In our study, we utilized two sets of HIV patient serum samples from different institutions, supporting that our results can be reproduced. We believe that these results significantly augmented the rigor of our findings.

There are concerns about the quality of the iPS-CMs since there is no cell imaging or molecular analyses. Figure 5 Supplement 1 images are of low quality and low resolution to assess cell quality. Overall the iPS-CM QC data is extremely sparse

We have now added the representative images of iPS-CMs to Figure 1- figure supplement 1A. Our group has used hiPS-CMs extensively in the past (PMID: 26439715). We also updated Fig 5 Supplement 1 with images with better resolution and added Fig 5 Supplement 2 with magnified images.

**Reviewer2 (Public Review):**
However, there are some topics that are not well-connected, and the rationale and hypothesis are not clearly defined beforehand, such as mitochondrial membrane potential, mitochondrial ROS, and angiogenic potential.

We modified the manuscript so the rationale and hypothesis of the study is clearly stated.

As the hiPSC cardiomyocytes are treated with various reagents to measure diastolic dysfunction, it is important to confirm whether the treatment time and dose used were sufficient to exert a functional effect. Dose and time-dependent experiments are essential, or at least sufficient citations should be provided for selecting the dose for IFN and TNF.

We used previous publications for the dosages of the drugs used in our paper (1-4).

After IFN and TNF treatment, determining the expression levels of molecular markers of DD/HFpEF is crucial. Again, if sufficient evidence is available, it can be cited.

We have included a section in the discussion to address this issue. Briefly, Lee et al. reported that 25 ng/ml TNFa induces a longer decay of calcium transient and a decrease in sarcoplasmic ATPase (SERCA) expression in rabbit cardiomyocytes from pulmonary vein (PMID 17383682). The prolonged Cadecay time in hiPS-CM with the drug administration may be due to a decrease in SERCA expression and impaired Ca-uptake into sarcoplasmic reticulum.

The Methods section describes TMRE colocalization and immunofluorescence, but no images are provided.

We have performed immunofluorescence of hiPSC-CM with TMRE for the quantification of mitochondrial membrane potential (MMP).

The concentration of TNF and IFN in patients is critical, which was acknowledged and discussed as a limitation of the study by the authors. Authors should consider this aspect, and if not feasible, clinical reports should be cited to provide a rough estimation of their concentration.

Thank you for this comment. A new section detailing the points brought up by the Reviewer is now added to discussion.

**Recommendation for the authors**:
**Reviewer #1 (Recommendation for the authors):**
I suggest a more comprehensive analysis of diastolic function including biomechanical studies of contraction and diastolic function. I suggest increasing the sample #'s, getting a better characterziation of the cardiomyocytes, their expression profiles, and maturation state. The team should dig more deeply into potential molecular mechanisms of the calcium transient changes. Are there changes in SERCA or other SR factors' phosphorylation state or other molecular explanations for the observed changes? I would remove the serum treatment experiments as they distract since they didn't show differences. These are a few of the suggestions I would have for the team.

Our system for measurement of Ca-transient unfortunately does not allow to obtain data on the cellular biomechanical property. We modified the manuscript so the results are not overstated and that the interpretation is correct. Since each dot in bar-graphs for Ca-transient data represents the average of signals generated from 75 K cells, we believe that the number of cells analyzed was sufficient for the analyses. Although it is not conclusive, previous reports suggested induction of SERCA2A expression by TNF-α treatment in isolated cardiomyocytes, suggesting that the mechanism underlying the prolonged calcium decay time in our model may be due to changes in SERCA levels. We included the data from human serum samples from HIV patients since they provide a platform to assess the effects of HIV patient serum on. We believe that these data convey a significant progress understanding the process of myocardial dysfunction in HIV patients.

References

Amirayan-Chevillard, N., Tissot-Dupont, H., Capo, C., Brunet, C., Dignat-George, F., Obadia, Y., Gallais, H., and Mege, J. L. (2000) Impact of highly active anti-retroviral therapy (HAART) on cytokine production and monocyte subsets in HIV-infected patients. Clinical and experimental immunology 120, 107-112

Fraietta, J. A., Mueller, Y. M., Yang, G., Boesteanu, A. C., Gracias, D. T., Do, D. H., Hope, J. L., Kathuria, N., McGettigan, S. E., Lewis, M. G., Giavedoni, L. D., Jacobson, J. M., and Katsikis, P. D. (2013) Type I interferon upregulates Bak and contributes to T cell loss during human immunodeficiency virus (HIV) infection. PLoS Pathog 9, e1003658

Lau, S. L., Yuen, M. L., Kou, C. Y., Au, K. W., Zhou, J., and Tsui, S. K. (2012) Interferons induce the expression of IFITM1 and IFITM3 and suppress the proliferation of rat neonatal cardiomyocytes. Journal of cellular biochemistry 113, 841-847

Stone, S. F., Price, P., Keane, N. M., Murray, R. J., and French, M. A. (2002) Levels of IL-6 and soluble IL-6 receptor are increased in HIV patients with a history of immune restoration disease after HAART. HIV Med 3, 21-27